# 4-Hydroxyphenyllactic Acid in Cerebrospinal Fluid as a Possible Marker of Post-Neurosurgical Meningitis: Retrospective Study

**DOI:** 10.3390/jpm12030399

**Published:** 2022-03-04

**Authors:** Alisa K. Pautova, Anastasiia Yu. Meglei, Ekaterina A. Chernevskaya, Irina A. Alexandrova, Natalia V. Beloborodova

**Affiliations:** 1Federal Research and Clinical Center of Intensive Care Medicine and Rehabilitology, 25-2 Petrovka, 107031 Moscow, Russia; amegley@fnkcrr.ru (A.Y.M.); echernevskaya@fnkcrr.ru (E.A.C.); nvbeloborodova@yandex.ru (N.V.B.); 2National Medical Research Center for Neurosurgery named after Academician N.N. Burdenko, 4th Tverskaya-Yamskaya, 16, 125047 Moscow, Russia; ialexandrova@nsi.ru

**Keywords:** microbial metabolites, interleukin-6, neurosurgical patients, nosocomial infection

## Abstract

The search for new potential biomarkers for the diagnostics of post-neurosurgical bacterial meningitis is required because of the difficulties in its early verification using results of the routine laboratory and biochemical analyses of the cerebrospinal fluid (CSF). The goal of the study was to determine the contents of the aromatic metabolites and biomarkers in the CSF samples of the post-neurosurgical patients (*n* = 82) and their potential diagnostical significance for the evaluation of the risk of post-neurosurgical meningitis. Patients with signs of post-neurosurgical meningitis (*n* = 30) had lower median values of glucose and higher values of cell count, neutrophils, lactate, protein, 3-(4-hydroxyphenyl)lactic acid (*p*-HPhLA), and interleukin-6 (IL-6) than patients without signs of post-neurosurgical meningitis (*n* = 52). ROC analysis for IL-6 and *p*-HPhLA resulted in 0.785 and 0.734 values of the area under the ROC curve, with sensitivity 96.30 and 66.67%; specificity 54.17 and 82.69%, respectively. IL-6 should be considered as a non-specific biomarker, in contrast to the microbial metabolite *p*-HPhLA. If the concentration of *p*-HPhLA was more or equal to 0.9 µmol/L, the risk of bacterial complications was 9.6 times higher. *p*-HPhLA is a promising marker for the prognosis of post-neurosurgical meningitis, and its determination on a larger group of post-neurosurgical patients can subsequently prove its diagnostic significance for the verification of CNS infections.

## 1. Introduction

The diagnostics of bacterial complications in patients of neurosurgical units is an important issue as these patients could be compromised by hospital-acquired or commensal bacteria, and they could have non-infectious complications that have similar symptoms, such as fever or decreased consciousness. Bacterial meningitis develops in post-neurosurgical patients (post-neurosurgical meningitis) after craniectomy, craniotomy (0.8–1.5% [1] and 0.3–8.6% [2] of patients), placement of internal (4–17% of patients [1,2]) or external (8% [1] and 0–22% [2] of patients) ventricular and lumbar (0.8–7% of patients [1,2]) catheters [2], lumbar puncture, intrathecal infusions of medications, spinal anesthesia, complicated head trauma (1.4% of patients), and sometimes infection in patients with hospital-acquired bacteremia [1].

Clinical signs of post-neurosurgical meningitis, such as headache, fever and decreased consciousness, are non-specific and could be caused by other reasons including non-infectious ones. Thus, a lumbar puncture after the CT scan is performed to obtain CSF samples for the laboratory, biochemical and microbiological analyses. The main diagnostic criterium of post-neurosurgical meningitis is the positive CSF Gram staining or microbiological culture, which is often false negative because of the antimicrobial therapy prescribed before or after the neurosurgery or for other reasons. Thus, laboratory parameters of CSF are used to judge the presence of post-neurosurgical meningitis. The most common parameters are neutrophilic pleocytosis (neutrophil predominance more than 80%), raised CSF protein, and hypoglycorrhachia, which are non-specific criteria and may indicate non-infectious diseases—for example, tumors [2,3,4,5]. CSF lactate with a cut-off value of >4 mmol/L [6] and elevated CSF procalcitonin [7] are also known to be possible biomarkers of post-neurosurgical meningitis and are used together with other biochemical parameters. The CSF levels of biomarkers, which are usually detected in serum, such as interleukin-6 (IL-6) as an indicator of inflammation process [8], neuron-specific enolase (NSE) [9] and S100 protein [10] as indicators of the brain cell damage, are of great interest to study different pathologies, including post-neurosurgical meningitis.

Metabolomics is actively used in searching for new biomarkers of bacterial meningitis. In a recent study, the authors compared concentrations of about 200 metabolites in CSF samples using liquid chromatography coupled to tandem mass spectrometry (HPLC-MS/MS) from patients with bacterial or viral meningitis and non-inflamed controls. The CSF levels of phosphatidylcholines (µmol/L levels) were significantly higher in patients with bacterial meningitis than in controls, while its serum concentrations remained relatively unchanged and had higher sensitivity and negative predictive values than the CSF lactate or cell count [11]. Phosphatidylcholines are common constituents of cell membranes and also occur in freeform in body fluids including blood and CSF, thus, this group of compounds is not specific for bacteria. In contrast, an interesting potential specific biomarker of bacterial meningitis is muramic acid, which is a constituent of the peptidoglycan backbones of Gram-positive and Gram-negative bacteria. This metabolite was not detected in humans without meningitis, but its levels in patients with pneumococcal meningitis ranged from 0.03 to 15 µmol/L. Its concentrations were detected using gas chromatography coupled to tandem mass spectrometry (GC-MS/MS) [12].

Some aromatic metabolites of tyrosine and phenylalanine were revealed to increase in the serum of patients with severe infectious diseases such as sepsis [13] and septic shock caused by community-acquired pneumonia [14], acute surgical diseases of abdominal organs [15], and in chronically critically ill patients [16,17]. Phenyllactic, 4-hydroxyphenylacetic and 4-hydroxyphenyllactic (*p*-HPhLA) acids are tyrosine and phenylalanine metabolites of microbial origin [18], and their serum concentrations correlate with the severity of the patient’s condition (presence of septic shock, SOFA and APACHE II scores), lactate, procalcitonin, and homovanillic acid (HVA) levels. These aromatic metabolites, together with other related compounds (benzoic (BA), phenylpropionic, 4-hydroxybenzoic acids, and HVA), were detected at µmol/L levels using GC-MS [19] and HPLC-MS/MS [20] in the CSF samples of patients with different CNS pathologies, but their diagnostic significance for the CNS diseases has not yet been revealed.

Thus, the goal of the study was to determine the contents of the aromatic metabolites and biomarkers in the CSF samples of the post-neurosurgical patients and their potential diagnostical significance for the evaluation of the risk of post-neurosurgical meningitis.

## 2. Materials and Methods

### 2.1. Biological Samples

The residues of CSF samples after routine laboratory analyses from post-neurosurgical patients (*n* = 82) were collected and frozen at −30 °C in the Federal Research and Clinical Center of Intensive Care Medicine and Rehabilitology (Moscow, Russia) as it was previously described [19]. The approval of the local ethics committee was obtained (N 2/19/2). The CSF samples were collected at the time of the suspected post-neurosurgical meningitis. Concentrations of aromatic metabolites (phenylpropionic, phenyllactic, 4-hydroxybenzoic, 4-hydroxyphenylacetic acids, BA, HVA, and *p*-HPhLA) were measured using GC-MS obtained from Thermo Scientific (Trace GC 1310 gas chromatograph and ISQ LT mass spectrometer, Thermo Electron Corporation, Waltham, MA, USA) [19]. Biomarkers (IL-6, S100 protein, and NSE) were analyzed using electrochemiluminescence (Cobas e411, Roche, Basel, Switzerland); 5-hydroxyindolacetic acid (5HIAA) was detected using Cloud-Clone kit (Houston, TX, USA) for the enzyme-linked immunosorbent assay (Multiscan, Thermo Electron Corporation, Waltham, MA, USA).

### 2.2. Patients

Information about the main disease and complications, clinical signs of infection, results of the CT scan, antimicrobial treatment, and laboratory analyses of the serum and CSF—including examination for cell count, protein, lactate, glucose levels, and microbiological culture from post-neurosurgical patients—was obtained from medical documentation retrospectively. Patients with a benign tumor (*n* = 42), malignant tumor (*n* = 18), stroke (*n* = 8), intracranial injury (*n* = 5), cyst (*n* = 4), hydrocephalus (*n* = 3), bacterial meningitis (*n* = 2) were divided into two groups. The first group included patients with signs of post-neurosurgical meningitis. The patients of the second group did not have sufficient signs of bacterial meningitis. The criteria of post-neurosurgical meningitis are listed below.

Major criteria:-Information on proved or suspected bacterial meningitis in the medical records according to the clinical signs;-Positive bacterial CSF culture;-CSF leukocyte count more than 300 cells/mm^3^ with the relative number of neutrophils more than 80%.

Minor criteria (for CSF):-Glucose level less than 2.7 mmol/L;-Lactate level more than 4 mmol/L;-Protein level more than 1 g/L.

Other non-specific criteria:-Hyperthermia/hypothermia;-The presence of the draining devices.

If there were no major criteria, the patient was classified in the group with signs of post-neurosurgical meningitis when there were 3 or more signs of minor criteria. In total, there were 30 from 82 patients with signs of post-neurosurgical meningitis (Table 1).

### 2.3. Statistical Analysis

One CSF sample from each patient was used for statistical analysis. The accumulation of data and the primary analysis were carried out in Microsoft Office Excel 2019. Statistical data analysis was carried out using the IBM SPSS Statistics 25.0 application package and online calculators (https://medstatistic.ru/calculators/calchi.html (accessed on 9 December 2021)).

To test for normality, we used either the Shapiro–Wilks test for small samples or the Kolmogorov–Smirnov test with the Lilliefors correction for larger samples. The equality of variances was checked using the Levene’s test. According to the results of tests, it was revealed that all of the quantitative indicators of the parametric comparison criteria are inapplicable due to the small number of outcomes; therefore, the comparative intergroup analysis was carried out using the nonparametric Mann–Whitney U-test (with the values of statistics tests U and Z); for multiple pairwise comparisons, the Bonferroni correction was used. Correlation analysis was carried out using Spearman’s nonparametric correlation coefficient. Values are reported as medians and interquartile ranges. The significance level was chosen equal to 0.05 (a different value in case of using the Bonferroni correction). To assess the quality of various quantitative indicators, ROC analysis with the area under the ROC curve was used.

Additionally, the following parameters were evaluated: precision or positive predictive value, negative predictive value, accuracy, and odds ratio:$$\mathrm{Positive}\;\mathrm{predictive}\;\mathrm{value}=\;\frac{\mathrm{Se}\times P}{\mathrm{Se}\times P+\left(1-\mathrm{Sp}\right)\times \left(1-P\right)};$$
$$\mathrm{Negative}\;\mathrm{predictive}\;\mathrm{value}=\;\frac{\mathrm{Se}\times \left(1-P\right)}{\left(1-\mathrm{Se}\right)\times P+\mathrm{Sp}\times \left(1-P\right)};$$
where Se—Sensitivity, Sp—Specificity, and P—Prevalence.

The main criteria for the success of the predictor are a high area under the ROC curve and the lower bound of the confident interval being not less than 50% for the odds ratio, sensitivity, specificity, positive and negative predictive values, and accuracy.

## 3. Results

### 3.1. Characteristics of the Patients with Signs of Post-Neurosurgical Meningitis

A smaller part (*n* = 12) of the group of patients with signs of post-neurosurgical meningitis (*n* = 30) was the patients (Table 1) who had information about the presence of the proved/suspected bacterial meningitis as an infectious complication in medical documentation (*n* = 8) or had positive bacterial CSF culture (*n* = 6). To classify the majority of patients in this group (*n* = 18), non-specific criteria for post-neurosurgical meningitis were used, which should be considered as potentially false results, as they could be caused by the main diagnoses of these patients (2/3 of the patients had tumors as the main diagnosis). This situation is the limitation of our study—although, this is a well-known problem of the post-neurosurgical meningitis diagnostics described in different studies (see Section 1). The primary diagnosis and the draining devices were not specific for the group of the patients with signs of post-neurosurgical meningitis in our study.

### 3.2. Aromatic Metabolites and Biomarkers in the CSF

The results of laboratory tests (the cell count, neutrophil content, levels of lactate, protein, and glucose) of the CSF for two groups of patients were obtained from medical records and are accumulated in Table 2. All residues of the CSF samples (*n* = 82) were analyzed using GC-MS for a number of aromatic metabolites [19]. They are phenylpropionic, phenyllactic, 4-hydroxybenzoic, 4-hydroxyphenylacetic acids, BA, HVA, and *p*-HPhLA. Most of these acids (phenylpropionic, phenyllactic, 4-hydroxybenzoic, and 4-hydroxyphenylacetic acids) could not be statistically evaluated as their concentrations were less than the limit of the quantitation in most samples. The concentrations of BA, HVA, and *p*-HPhLA together with biomarkers (IL-6, S100, NSE, and 5HIAA) are demonstrated in Table 2. IL-6 and S100 were measured in 75 CSF samples, 5HIAA and NSE, as well as in 74 and 73 CSF samples, respectively.

According to the data in Table 2, the patients with signs of post-neurosurgical meningitis had higher median values of leukocyte count, neutrophils, lactate, protein, *p*-HPhLA (Figure 1a), and IL-6 (Figure 1b), and lower values of glucose.

The ROC analysis was carried out for the parameters, which were significantly different in two groups of patients. The values of the leukocyte count, neutrophils, lactate, protein, and glucose were used in combination as the non-specific criteria for the presence of post-neurosurgical meningitis (see Section 2.2), and their statistical significance as the predictors was possible to evaluate correctly using ROC analysis. The ROC analysis results for *p*-HPhLA and IL-6 could be statistically evaluated and are demonstrated in Table 3. The areas under the ROC curve for IL-6 and *p*-HPhLA are within 0.7–0.8 and can be considered as acceptable [21]. The cut-off values were evaluated for IL-6 and *p*-HPhLA. If the concentration of IL-6 was more or equal to 270 pg/mL, the risk of bacterial complications was 30.7 times higher; if the concentration of *p*-HPhLA was more or equal to 0.9 µmol/L, the risk of bacterial complications was 9.6 times higher.

*p*-HPhLA is known to correlate with the severity of the bacterial complications and serum levels of some biomarkers in critically ill patients (Section 1), and we analyzed the information in the medical records that could indicate the presence of systemic inflammation. The following parameters were chosen (in serum): the total leukocyte count and C-reactive protein. The criteria of the systemic inflammation were both the total leukocyte count being less than 4 × 10^9^/L or more than 9 × 10^9^/L and C-reactive protein being more than 7 mg/L. In total, there were groups of patients with (*n* = 31) and without (*n* = 25) signs of systemic inflammation, and there was a group of patients without information about both the total leukocyte count and C-reactive protein in the serum samples collected on the same day of the CSF analysis (*n* = 26). The ROC analysis results for the *p*-HPhLA and systemic inflammation are demonstrated in Table 3, and most of parameters are similar to those for post-neurosurgical meningitis.

In addition, *p*-HPhLA is known to correlate with the serum lactate, and we revealed a moderately positive statistically significant correlation between the CSF levels of *p*-HPhLA and lactate (*r =* 0.55, *p =* 0.01) [22]. IL-6 correlated with the most common CSF parameters such as cell count, neutrophils, lactate, and protein (*r >* 0.5, *p =* 0.01). Other correlation coefficients are demonstrated in Figure 2.

## 4. Discussion

In this study, we determined the contents of aromatic metabolites and biomarkers in the CSF samples of post-neurosurgical patients to reveal their potential diagnostical significance for the evaluation of the risk of post-neurosurgical meningitis. Post-neurosurgical meningitis may develop in post-neurosurgical patients regardless of the main disease (tumors, intracranial injure, stroke, etc.). This fact is the reason for the heterogeneity of the patients’ cohort included in the study, and it is diagnostically important to identify a marker of post-neurosurgical meningitis, which will not depend on the main disease but will indicate an infectious process.

### 4.1. Biomarkers

Several biomarkers (IL-6, S100, NSE, and 5HIAA) were studied in our research, and only one of them (IL-6) demonstrated potential diagnostic significance for the prognosis of post-neurosurgical meningitis. Serum IL-6 is used as a diagnostic and prognostic biomarker for inflammation, autoimmune disorders, cancer [8,23], cardiovascular diseases [24], and infection [25].

IL-6 is one of the most important cytokines involved in the inflammatory response in the CNS, where it is produced by endothelial cells, astrocytes and glial cells as a response to various injuries and is also involved in neurogenesis. It induces the synthesis of acute-phase proteins and contributes to blood–brain barrier damage [26]. Several studies reported that the CSF levels of IL-6 were much higher than the serum levels in patients with subarachnoid hemorrhage, chronic inflammatory disease, seizures, and post-traumatic stress disorder [27,28,29,30]. The lack of a clear relationship between IL-6 in CSF and plasma suggests that the plasma concentrations of inflammatory markers cannot provide useful information about CNS inflammation.

In our study, the IL-6 cut-off value (270 pg/mL) was lower compared with other studies. In a prospective study [31] of the CSF samples of neurosurgical patients (*n* = 75) with an external ventricular drainage, which had been inserted predominantly because of poor-grade subarachnoid hemorrhage, the intrathecal IL-6 concentrations correlated with the clinical course and ventriculostomy-related infection incidence. The predictive value of CSF IL-6 in patients with an external ventricular drainage for the diagnosis of CNS infection prior to clinically manifesting meningitis was 2700 pg/mL. In another study [32], the level of IL-6 in the CSF of patients with a diagnosis of purulent bacterial meningitis with a more severe course reached 392 ± 199 pg/mL. Retrospective data [33] from patients with traumatic brain injury (*n* = 40) and external ventricular drain-associated ventriculitis showed that the CSF IL-6 levels higher than the threshold of 4064 pg/mL were significantly associated with the probability of ventriculitis. In another prospective study [29], all patients with proven bacterial meningitis had CSF IL-6 more than 500 pg/mL. In a retrospective study of the CSF samples of patients with clinical signs of meningitis (*n* = 106) [34], IL-6 was studied for early differentiation between aseptic and bacterial meningitis with a cut-off value of 1418 pg/mL (95.5% sensitivity and 77.5% specificity) or 15,060 pg/mL (63.6% sensitivity and 96.7% specificity) for thediagnosis of bacterial meningitis. All kits in the studies [29,31,32,34] and the test method used in our study have been standardized against the National Institute for Biological Standards and Control’s (NIBSC) first international standard (89/548) with close limits of detection linearity; thus, the differences in the IL-6 concentration should be considered unrelated to the test systems.

Patients with signs of post-neurosurgical meningitis manifested statistically significant higher median values of IL-6 (up to a 10-fold increase in median, Table 2). However, the specificity of IL-6 was low, and the value of the lower bound of the CI was less than 50%, i.e., 54.17% [95% CI 39.17–68.63] (Table 3). Thus, the elevated level of IL-6 in the CSF is not a specific and reliable diagnostic and prognostic biomarker in the differential diagnosis of meningitis, especially in patients without sufficient signs of bacterial meningitis.

According to Figure 1a, about 50% of patients without sufficient signs of bacterial meningitis had elevated CSF IL-6 (≥270 pg/mL), and most of them had malignant or benign tumors as the main diagnoses. Elevated levels of IL-6 in the CNS have been found in inflammatory CNS diseases, such as idiopathic transverse myelitis [35], and neuromyelitis optica [36], cancer, and cysts [37,38] (median CSF IL-6 654.3 ± 247.9, 281, 22.87  ±  2.62, and 594.2  ±  191.8 pg/mL, respectively). Thus, the main diagnosis could be the reason for the elevated levels of IL-6 but not for post-neurosurgical meningitis.

### 4.2. Aromatic Metabolites

Some aromatic metabolites in CSF were of great interest in our study as they could be considered as specific bacterial biomarkers, especially one of them, *p*-HPhLA. The main assumption in our study is that the *p*-HPhLA concentration is associated with the presence of CSF infection in post-neurosurgical patients. Patients with signs of post-neurosurgical meningitis (*n* = 30) manifested 3 times higher median values of *p*-HPhLA (Table 2) than those without sufficient signs of post-neurosurgical meningitis (*n* = 52). The CSF levels of *p*-HPhLA in patients with signs of post-neurosurgical meningitis (*n* = 30) were higher than the cut-off value (0.9 µmol/L) in 20 cases (Figure 1b), which is consistent with the assumption that increased *p*-HPhLA levels are associated with the CNS infection. In 10 cases, *p*-HPhLA was detected in lower concentrations than the cut-off value. This could be explained by the wrong classification of patients into the group of patients with signs of post-neurosurgical meningitis because of the absence of a confirmed diagnosis by the positive microbiological culture in most cases (see Section 3.1). Thus, we used the recommended non-specific criteria (cell count, neutrophils, lactate, protein, and glucose levels) for the classification and their abnormal levels could be explained by the main diagnoses [39,40,41,42]: 7 of 10 patients with low level of *p*-HPhLA had different types of tumors, others had cysts. However, the *p*-HPhLA levels in patients with different types of tumors have not been studied before and are of great interest for the further research.

Special attention should be paid to the patients (*n* = 9) without sufficient signs of meningitis (Figure 1b) who had *p*-HPhLA levels higher than the cut-off value (0.9 µmol/L). Seven of them had at least signs of systemic inflammation in serum (C-reactive protein more than 7 mg/L and leukocyte count more than 1 × 10^10^/L); for two patients, there was no information about the serum levels of C-reactive protein and leukocyte count in the serum samples collected on the same day of the CSF analysis.

The obtained results could be summarized to a hypothesis about the mechanisms of the *p*-HPhLA accumulation in CSF (Figure 3). If there is no systemic inflammation in the body, aromatic metabolites of microbial origin (in particular, *p*-HPhLA) enter the bloodstream from the gut at constant low concentrations, as demonstrated in serum samples of healthy controls [43]. If there is a systemic inflammation in the post-neurosurgical patients with a locus of infection outside of CNS, we should expect elevated concentrations of *p*-HPhLA in the serum by analogy with the critically ill patients as in previous studies (see Section 1). *p*-HPhLA is a polar low-molecular-weight aromatic metabolite, and we assume that it can penetrate the blood–brain barrier with increased permeability, resulting in elevated CSF levels of *p*-HPhLA. Such situation was illustrated by our results in patients without sufficient signs of meningitis but with signs of systemic inflammation in serum (*n* = 9) and by the similar results of the ROC analysis of the CSF level of *p*-HPhLA and inflammation in CSF and serum (Table 3).

In the case of confirmed CNS infection, we should expect elevated *p*-HPhLA levels in CSF and consider it as a microbial indicator of post-neurosurgical meningitis.

Another point of the discussion is the ratio between serum and CSF concentration of *p*-HPhLA. Recent studies revealed that aromatic metabolites were detected in the CSF samples in lower concentrations than in serum. Similar correlations between concentrations of phenylpropionic, 4-hydroxybenzoic, phenyllactic, 4-hydroxyphenylacetic acids, BA, HVA and *p*-HPhLA in both biological fluids were revealed [19], which may indirectly indicate that low-molecular-weight aromatic metabolites can penetrate the damaged blood–brain barrier. Thus, we should expect the concentration of *p*-HPhLA to be higher in CSF than in serum in the case of infection locus in CNS; CSF concentration of *p*-HPhLA to be lower than in serum in the case of infection locus out of CNS. This hypothesis should be checked by the analysis of the serum and CSF samples taken at the same time from a large group of patients. The first results of the simultaneous analysis of CSF and serum in a small group of five patients demonstrated that the ratio between the serum and CSF concentration of *p*-HPhLA can be different [19].

## 5. Conclusions

The diagnostics of post-neurosurgical meningitis as an inflammation complication in post-neurosurgical patients is an important and difficult problem. The absence of positive CSF Gram staining or microbiological culture in most cases of post-neurosurgical meningitis forces the use of non-specific criteria, such as cell count, neutrophils, lactate, protein, and glucose levels, of which values could be different from the norm because of the main disease or other non-infectious reasons. Thus, the search for new and specific criteria of nosocomial meningitis is an actual issue. In our study, we evaluated the concentrations of different biomarkers (IL-6, NSE, S100, and 5HIAA) and some aromatic metabolites, including those of microbial origin, in patients with and without signs of post-neurosurgical meningitis. The results of our retrospective study demonstrated that IL-6 is a non-specific biomarker and its elevated concentrations in patients with signs of post-neurosurgical meningitis could be due to both meningitis and main disease. At the same time, *p*-HPhLA, an aromatic metabolite of microbial origin, can be considered as a specific criterion of post-neurosurgical meningitis. Despite promising statistical results for *p*-HPhLA (area under the ROC curve and other parameters), the main limitation is the value of the lower bound of the confident interval for sensitivity (less than 50%), which may be explained by the insufficient sample size. Further studies on a larger group of patients may improve the statistical results. Additionally, the analysis of *p*-HPhLA in serum and CSF samples taken simultaneously is needed and will reveal its pathophysiological and diagnostic significance for the verification of CNS infections.

## Figures and Tables

**Figure 1 jpm-12-00399-f001:**
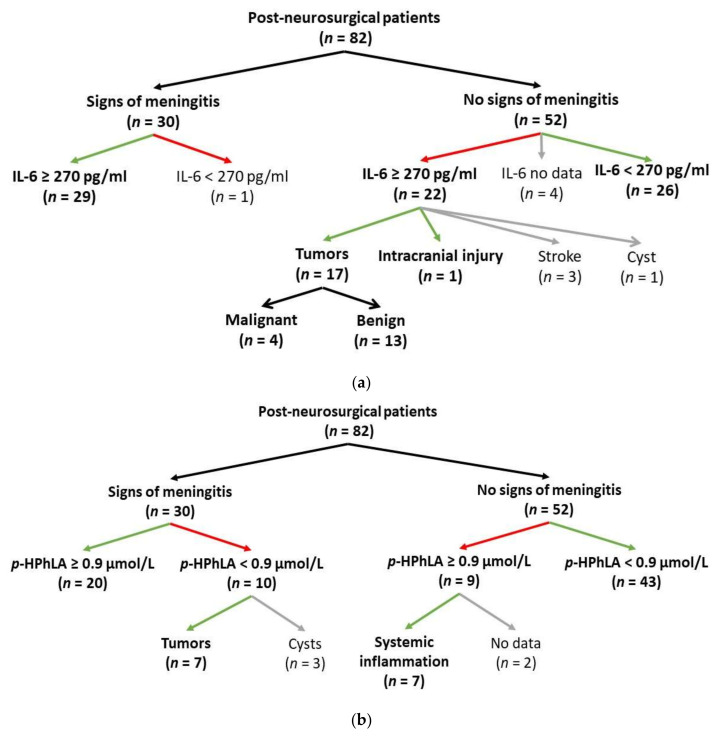
The CSF levels of IL-6 (**a**) and *p*-HPhLA (**b**) in patients with (*n* = 30) and without (*n* = 52) signs of post-neurosurgical meningitis.

**Figure 2 jpm-12-00399-f002:**
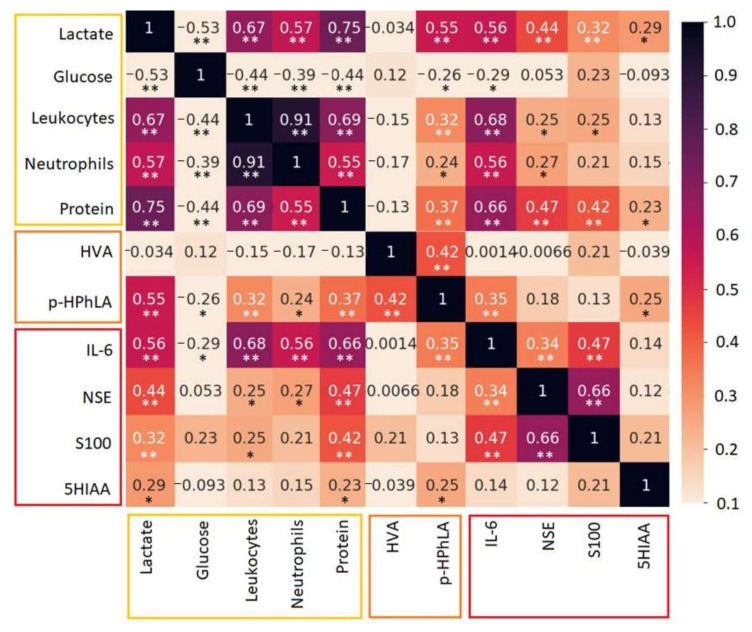
Spearman’s rank correlation coefficients for the different parameters in CSF: clinical and biochemical parameters (lactate, glucose, leukocytes, neutrophils, and protein), aromatic metabolites (HVA and *p*-HPhLA), and biomarkers (IL-6, NSE, S100, and 5HIAA). Negative correlations for glucose are not colored. * The correlation is significant at *p* = 0.05 (2-tailed); ** the correlation is significant at *p* = 0.01 (2-tailed).

**Figure 3 jpm-12-00399-f003:**
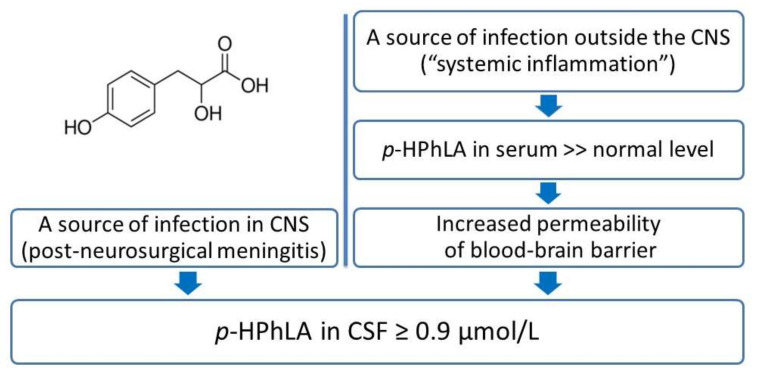
The potential infectious sources of increased *p*-HPhLA levels in CSF of post-neurosurgical patients.

**Table 1 jpm-12-00399-t001:** Characteristics of the patients with and without signs of post-neurosurgical meningitis.

Parameter	Patients with Signs of the Post-Neurosurgical Meningitis (*n* = 30)	Patients without Sufficient Signs of the Bacterial Meningitis (*n* = 52)
Sex, male/female	*n* = 18/*n* = 12	*n* = 22/*n* = 30
Primary diagnosis	benign tumor (*n* = 10)	benign tumor (*n* = 32)
	malignant tumor (*n* = 10)	malignant tumor (*n* = 8)
	stroke (*n* = 1)	stroke (*n* = 7)
	intracranial injury (*n* = 3)	intracranial injury (*n* = 2)
	cyst (*n* = 3)	cyst (*n* = 1)
	hydrocephalus (*n* = 2)	hydrocephalus (*n* = 1)
	bacterial meningitis (*n* = 1)	bacterial meningitis (*n* = 1) ^1^
Non-survived patients	*n* = 2	*n* = 1
Proved/suspected bacterial meningitis as an infectious complication	*n* = 6/*n* = 2	0
Positive bacterial CSF culture	*n* = 6	0
CSF leukocyte count more/less/no data than 300 cells/mm^3^	*n* = 25/*n* = 5/*n* = 0	*n* = 4/*n* = 44/*n* = 4
CSF relative number of the neutrophils more/less/no data than 80%	*n* = 26/*n* = 2/*n* = 2	*n* = 9/*n* = 33/*n* = 10
CSF glucose level more/less/no data than 2.7 mmol/L	*n* = 12/*n* = 17/*n* = 1	*n* = 46/*n* = 4/*n* = 2
CSF lactate level more/less/no data than 4 mmol/L	*n* = 18/*n* = 10/*n* = 2	*n* = 5/*n* = 40/*n* = 7
CSF protein level more/less/no data than 1.0 g/L	*n* = 25/*n* = 4/*n* = 1	*n* = 18/*n* = 33/*n* = 1
Draining devices	*n* = 12	*n* = 13
Hyperthermia	*n* = 9	*n* = 3

^1^ The CSF sample was collected after 2 months of acute bacterial meningitis and where the patient had no clinical signs of bacterial meningitis.

**Table 2 jpm-12-00399-t002:** The results of the laboratory tests and the concentrations of aromatic metabolites and biomarkers in the CSF samples of the post-neurosurgical patients (*n* = 82), including the results of the Mann–Whitney U-test. Data are presented as median [interquartile range 25–75%], minimum–maximum values.

Parameter	Patients with Signs of Post-Neurosurgical Meningitis (*n* = 30)	Patients without Sufficient Signs of the Bacterial Meningitis (*n* = 52)	The Mann–Whitney U-Test, *p*-Value
Age, years	34 [14.5–53], 0.7–67	33 [10–54], 2–91	0.365
Leukocyte count, cells/mm^3^	936 [504–2688], 38–20488	17 [7–72], 1–672	<0.0001 ***
Neutrophils, %	96 [88–98], 33–100	23 [7–66], 1–100	<0.0001 ***
Glucose, mmol/L	2.2 [0.8–3.2], 0.2–5.2	3.3 [2.9–4.1], 1.7–7.9	<0.0001 ***
Lactate, mmol/L	4.9 [3.6–6.2], 2.9–9.4	2.0 [1.7–3.2], 1.4–6.5	<0.0001 ***
Protein, g/L	3.2 [1.2–3.9], 0.4–22.8	0.6 [0.3–1.5], 0.1–3.4	<0.0001 ***
BA, µmol/L	0.7 [0 *–0.8], 0 *–1.0	0.8 [0.7–1.0], 0 *–2.4	0.127
HVA, µmol/L	0 * [0 *–0 *], 0 *–1.5	0 * [0 *–0.5], 0 *–3.3	0.311
*p*-HPhLA, µmol/L	1.1 [0.5–1.6], 0 *–6.6	0.4 [0 *–0.7], 0 *–5	0.00019 ***
5HIAA, ng/mL	5 [3–7], 2–400 **	5 [3–6], 2–400 **	0.631
IL-6, pg/mL	2678 [600–5000 **], 163–5000 **	227 [33–1825], 7–5000 **	0.00002 ***
S100, µg/L	2.9 [0.6–13.2], 0.1–39 **	3.5 [0.4–22.0], 0.2–39 **	0.788
NSE, ng/mL	5.9 [1.6–32.6], 0.2–2496	2.4 [1.4–8.8], 0.5–370	0.122

* The concentrations are below the lower limit of quantitation (0.7 µmol/L for BA, 0.4 µmol/L for HVA and *p*-HPhLA); ** the concentrations are above the upper limit of quantitation (400 ng/mL for 5HIAA, 5000 pg/mL for IL-6, 39 µg/L for S100, and 370 ng/mL for NSE); *** statistically significant.

**Table 3 jpm-12-00399-t003:** The ROC analysis results for the *p*-HPhLA and IL-6 levels in CSF as the predictors of post-neurosurgical meningitis (CNS inflammation) and for the *p*-HPhLA in CSF as the predictor of systemic inflammation (in serum).

Parameter		CSF Level of IL-6	CSF Level of *p*-HPhLA	
		CNS Inflammation		Systemic Inflammation
Area		0.785	0.734	0.698
Standard Error		0.051	0.064	0.070
*p*-Value		0.001	0.001	0.011
Asymptotic 95% CI	Lower Bound	0.685	0.608	0.560
	Upper Bound	0.886	0.860	0.836
Cut-Off Value		270 pg/mL	0.9 µmol/L	0.9 µmol/L
Sensitivity [95% CI], %		96.30 [81.03–99.91]	66.67 [47.19–82.71]	48.39 [30.15–66.94]
Specificity [95% CI], %		54.17 [39.17–68.63]	82.69 [69.67–91.77]	76.00 [54.87–90.64]
Positive Predictive Value [95% CI], %		54.17 [46.28–61.85]	68.97 [53.81–80.91]	71.43 [53.24–84.59]
Negative Predictive Value [95% CI], %		96.30 [78.87–99.45]	81.13 [71.83–87.88]	54.29 [44.18–64.05]
Accuracy [95% CI], %		69.33 [57.62–79.47]	76.83 [66.20–85.44]	60.71 [46.75–73.50]
Odds Ratio [95% CI]		30.7 [3.9–245.1]	9.6 [3.4–27.2]	3.0 [0.9–9.4]
Cut-Off Value		270 pg/mL	0.9 µmol/L	0.9 µmol/L

## Data Availability

Not applicable.

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
