# Peer review of "4-Hydroxyphenyllactic Acid in Cerebrospinal Fluid as a Possible Marker of Post-Neurosurgical Meningitis: Retrospective Study"

_jpm, 2022, doi:10.3390/jpm12030399_

Round 1

Reviewer 1 Report

The authors have described biomarkers for the diagnostics of the post-neurosurgical bacterial meningitis. Considering the values of biochemical parameters, they have reached the conclusion that the microbial metabolite p-HPhLA can be considered as specific to determine the status of patients. Comment that figures 2 and 3 I think should be in the results section.
When comparing IL-6 results with other studies, authors should consider the experiment and control values of the kit used. They have been based on clinical data, but they should describe the products together with the commercial house that was used for them, so perhaps a more precise comparison could be reached.

The study presented may be interesting as they reveal a new marker

Author Response

Dear Reviewer,

Thank you very much for your review and your appreciation of our manuscript.

We analyzed the methods for the IL-6 detection in the studies from the Discussion section.  We add the information about test systems and standard protocols in the lines 287-291: "All kits in studies [29,31,32,34] and the test method used in our study have been standardized against the National Institute for Biological Standards and Control’s (NIBSC) first international standard (89/548) with close limits of detection linearity, thus, the differences in the IL-6 concentration should be considered unrelated to the test systems."

Also we replace the Fig. 2 and 3 into the Results section.

Reviewer 2 Report

The results of  the research presented in the article " 4-Hydroxyphenyllactic Acid in Cerebrospinal Fluid as a Possible Marker of the Post-Neurosurgical Meningitis: Retrospective  Study, are interesting but as the authors  presented, further studies on a larger group of patients may improve the statistical results.

Author Response

Dear Reviewer,

Thank you very much for your review and your appreciation of our article.